# Response of the Endophytic Microbiome in *Cotinus coggygria* Roots to Verticillium Wilt Infection

**DOI:** 10.3390/jof10110792

**Published:** 2024-11-15

**Authors:** Yanli Cheng, Juan Zhao, Yayong Liu, Taotao Zhang, Tom Hsiang, Zhihe Yu, Wentao Qin

**Affiliations:** 1College of Life Sciences, Yangtze University, Jingzhou 434025, China; orangefromkaifeng@hotmail.com; 2Institute of Plant Protection, Beijing Key Laboratory of Environment Friendly Management on Fruit Diseases and Pests in North China, Beijing Academy of Agriculture and Forestry Sciences, Beijing 100097, China; zhaojuan119882@163.com (J.Z.); liuyayong528@163.com (Y.L.); ztt1024@163.com (T.Z.); 3School of Environmental Sciences, University of Guelph, Guelph, ON N1G 2W1, Canada; thsiang@uoguelph.ca

**Keywords:** *Cotinus coggygria*, *Verticillium dahliae*, illumina sequencing, endophytic microbiome, community composition

## Abstract

Verticillium wilt caused by *Verticillium dahliae* Kleb. is a lethal soil-borne fungal disease of *Cotinus coggygria*. The plant endophytic microbiome plays an important role in maintaining plant health and disease resistance, but it is unclear how the endophytic microbiome of *C. coggygria* roots varies in response to Verticillium wilt occurrence. In this study, the endophytic microbial diversity, community composition, dominant species, and co-occurrence network of *C. coggygria* under Verticillium wilt-affected and healthy conditions were assessed using Illumina sequencing. Compared with healthy plants, the bacterial alpha diversity indices of Verticillium wilt-affected plants decreased significantly, while the fungal alpha diversity indices showed obvious increases. The relative abundance of dominant taxa including Proteobacteria, Actinobacteriota, Ascomycota, and Basidiomycota at the phylum level, as well as Gammaproteobacteria, Thermoleophilia, Dothideomycetes, and Agaricomycetes at the class level, differed significantly between Verticillium wilt-affected and healthy plants. Co-occurrence networks revealed that the fungal network of Verticillium wilt-affected roots was denser with more negative interactions, which may be relevant to functional changes from reciprocity to competition in the microbial community, in response to *V. dahliae* infection. The results enhanced our understanding on the relationships between the endophytic microbiome and Verticillium wilt, which could provide information for the management of this disease in *C. coggygria*.

## 1. Introduction

*Cotinus coggygria* is a Eurasian shrub plant of the Anacardiaceae family, known as the “smoke tree”, with important economic, medicinal, and ornamental values [1]. As a common red leaf ornamental species, *C. coggygria* is widely planted, especially at some famous scenic spots such as the Fragrant Hills and Badaling in China. Lei [2] first reported Verticillium wilt on *C. coggygria* caused by *V. dahliae* Kleb. As a typical soil-borne fungal agent, *V. dahliae* has a strong soil viability and adaptability with a wide host range [3]. Thousands of *C. coggygria* plants have been removed every year due to the severe disease in Fragrant Hills Park, which seriously affected the capital garden landscape and ecological construction [4]. The typical symptoms of Verticillium wilt on *C. coggygria* comprise the yellowing, wilting, and curling of leaf surfaces; the withering and drying up of inflorescences; the discoloration of vascular tissues; retarded growth; the withering of individual branches; and, ultimately, the demise of the entire tree [5]. With increasing planting areas of *C. coggygria*, the damage has become increasingly more noticeable.

The long-term use of chemicals has been associated with negative effects in terms of environmental pollution and the development of pesticide resistance. The endophytic microbiome is thought to be closely linked to plant growth and health. The endophytic microbiome is a subset of the plant microbiome, including bacteria and fungi that colonize plant cells or between cells, without causing noticeable harm to plants [6,7]. Some endophytes show biological activities such as phosphorus dissolution, nitrogen fixation, and the production of indoleacetic acid or antimicrobial compounds, which may be beneficial for plants for their absorption and utilization of nutrients, the regulation of growth-related hormones, and resistance to pests and diseases [8]. Therefore, a deeper understanding of the relationships of the endophytic microbiomes and health status of host plants may allow for the improved management of plant diseases [9]. In this study, we aimed to analyze the variability in microbial diversity, community composition, dominant species, and co-occurrence networks of the root endophytic microbiome in response to Verticillium wilt occurrence, and to provide a theoretical foundation for the effective prevention and control of Verticillium wilt on *C. coggygria*.

## 2. Materials and Methods

### 2.1. Sampling Site and Collection

The *C. coggygria* plant samples were collected in a woodland, of about 400 m^2^ in area, at the northern slope of the Fragrant Hills Park (39°99′61′′ N and 116°19′50′′ E) in the Haidian district of Beijing, China, in August 2021. This region has a sub-humid warm temperate continental monsoon climate with an annual precipitation of approximately 600 mm [10]. All the *C. coggygria* var. pubescens were about 2.5 to 3 m in height, with a crown diameter of 2.5 to 3 m and a trunk diameter of 10 cm. Firstly, we investigated *C. coggygria* plants at the sampling site and graded the disease severity of Verticillium wilt from 0 to 4 [5]—Level 0: no wilting; Level 1: <5 leaves turn yellow or wilt; Level 2: 5–10 leaves turn yellow or wilt; Level 3: leaves on 2/3 stems turn yellow or wilt; Level 4: >85% of the leaves wilted, fell off, or died entirely. Secondly, we also detected the pathogen of selected plants using universal primers and specific primers for *V. dahliae* to determine whether it carries the pathogen of Verticillium wilt [5]. In this study, *C. coggygria* plants with a disease severity of Verticillium wilt rated as grade 3–4 and determined to be positive for pathogen detection were classified as Verticillium wilt-affected plants, and those without wilting symptoms and negative for pathogen detection were considered healthy plants. A total of 15 Verticillium wilt-affected and 15 healthy plants of *C. coggygria* var. pubescens were selected to obtain plant samples, with three replications and each replication consisting of five plants. After carefully digging out the plant fiber roots, the soil tightly adhering to the root surfaces was washed off with phosphate-buffered saline. The root samples were cut into 5 cm lengths, surface-sterilized in 75% ethanol for 2 min and 1% NaClO for 1 min, and finally washed three times with sterile H_2_O [11]. The Verticillium wilt-affected root samples (RP) and healthy samples (RH) were stored at −80 °C prior to DNA extraction.

### 2.2. DNA Extraction, PCR, and Illumina Metagenomic Sequencing

The genomic DNA was extracted using the FlaPure Plant DNA Extraction Kit (Beijing Genesand Biotech Co., Ltd., Beijing, China) according to the manufacturer’s protocols. The DNA concentration and purity were determined using the NanoDrop2000 UV–Visible spectrophotometer (Thermo Fisher Scientific, Wilmington, DE, USA). The V5–V7 hypervariable region of 16S rDNA from bacteria was amplified using primer pairs 799F (5′-AACMGGATTAGATACCCKG-3′) and 1193R (5′-ACGTCATCCCCACCTTCC-3′) [12], and the ITS1-ITS2 region of fungi was amplified with ITS1F (5′-CTTGGTCATTTAGAGGAAGTAA-3′) and ITS2R (5′-GCTGCGTTCTTCATCGATGC-3′) [13]. High-throughput sequencing was performed using the Illumina Sequencing platform at Majorbio Bio-Pharm Technology Co., Ltd. (Shanghai, China), and the raw data were uploaded to the NCBI Sequence Read Archive with the submission accession numbers SAMN40539537-40539542 and SAMN40539821-40539826 under BioProject ID PRJNA1089384 and ID PRJNA1089391 for the bacterial and fungal sequences, respectively.

### 2.3. Data Processing and Sequence Analysis

The original sequences were quality controlled, before being filtered using Flash version 1.2.11 [14]. Valid labels were clustered using Uparse version 11 [15] for ≥97% sequence similarity, and such labels were clustered in the same operational taxon (OTU). Mothur (version 1.30.2) [16] was used to analyze α-diversity indices. Principal component analysis (PCA) was performed based on an analysis of similarity (Anosim), and the coordinates were used to visualize differences in microbial community structure. LEfSe (Linear Discriminant Analysis Effect Size) software v 1.0 [17] was used to identify the differentially abundant OTUs among the samples for biomarker discovery. Co-occurrence networks were identified as statistically robust correlations in abundance among OTUs (Spearman’s correlation coefficient *ρ* > 0.8, *p* <  0.05), and network visualization was performed using Gephi 0.10.1 [18].

### 2.4. Data Statistical Analysis

Data were first analyzed using analysis of variance, as implemented in Excel 2016 software. When significant treatment effects were found (*p* < 0.05), means were separated by the test of least significant difference.

## 3. Results

### 3.1. Sequencing Data Analyses

A total of 1235 bacterial and 251 fungal OTUs were obtained for the endophytic microbiome of *C. coggygria* root samples, using the 97% similarity cut-off. Between Verticillium wilt-affected and healthy plants, 322 bacterial OTUs were shared, and there were 22 OTUs unique to Verticillium wilt-affected, and 891 OTUs specific to healthy plants (Appendix A). For fungal OTUs, 28 were shared, while 186 were specific to Verticillium wilt-affected, and 37 were specific to healthy plants (Appendix A). The results indicated that healthy roots had more unique bacterial OTUs, while diseased roots possessed more specific fungal OTUs.

### 3.2. Alpha Diversity Analysis of the Root Endophytic Microbiome

Alpha diversity analysis showed that the Sobs, Chao, Ace, Shannon, and Invsimpson indices of endophytic bacteria in Verticillium wilt-affected *C. coggygria* plants were significantly lower than those of healthy plants, indicating reduced richness, evenness, and diversity of endophytic bacteria in the diseased plants (Figure 1A and Appendix A). The Sobs, Chao, and Ace indices for endophytic fungi were notably increased in the diseased *C. coggygria* plants compared with the healthy plants. No significant differences were observed for the Shannon and Invsimpson indices for endophytic fungi between Verticillium wilt-affected and healthy roots (Figure 1B and Appendix A).

### 3.3. Community Composition of the Root Endophytic Microbiome

For bacterial communities, Proteobacteria, Actinobacteriota, Firmicutes, Myxococcota, and Chloroflexi were the dominant taxa at the phylum level (Figure 2A), and Alphaproteobacteria, Actinobacteria, Gammaproteobacteria, and Thermoleophilia were the primary species at the class level (Figure 2B). The relative abundance of Proteobacteria was significantly increased in the roots of Verticillium wilt-affected plants, compared with that of healthy roots; meanwhile, the relative abundance of Actinobacteriota obviously decreased in the roots of Verticillium wilt-affected plants compared to healthy roots. At the class level, the relative abundance of Gammaproteobacteria was higher in the roots of the diseased plants compared to healthy roots, and the relative abundance of Thermoleophilia was reduced distinctly compared with that of healthy roots (Appendix A).

Fungal classes Dothideomycetes (Ascomycota) and Agaricomycetes (Basidomycota) were the dominant taxa in the fungal community (Figure 2C,D). In addition, compared with healthy roots, the relative abundance of Dothideomycetes declined significantly, while that of Agaricomycetes was increased in the diseased roots (Appendix A).

### 3.4. Beta Diversity of the Root Endophytic Microbiome

There was a significant difference in microbial community structure between Verticillium wilt-affected and healthy plants (Figure 3). The principal component analysis (PCA), based on the Anoism analysis, explained over 81% of the variation in the bacterial communities (Figure 3A). The community structures of the root endophytic bacteria in Verticillium wilt-affected and healthy *C. coggygria* were clearly separated in the first axis (54.44%), indicating the significant difference in their community composition. The endophytic fungal community of Verticillium wilt-affected and healthy plants was differentiated in the first axis (35.01%) (Figure 3B), which suggested distinct differences in fungal community structure between the two groups.

### 3.5. Biomarker Species of the Root Endophytic Microbiome

LEfSe analysis revealed the biomarker species in the root endophytic microbial communities of the Verticillium wilt-affected and healthy *C. coggygria.* For bacterial communities (Figure 4), *Sphingomonas*, *Acidisoma*, and *Pseudomonas* were significantly enriched in the Verticillium wilt-affected roots, while *Myxococcota*, *Bacillus*, and *Actinobacteriota* were relatively and significantly greater in healthy roots. An analysis of fungal components showed that *Mortierella* and *Flagelloscypha* were relatively higher in diseased roots, while *Acrocalymma* and *Russula* were relatively more present in healthy roots (Figure 5). Linear Discriminant Analysis (LDA) revealed that root endophytic microbial taxa significantly differed between the Verticillium wilt-affected and healthy root samples (Appendix A).

The relative abundance of the dominant microbial taxa between the Verticillium wilt-affected and healthy plant roots was also compared based on heatmap analysis. The results showed that the relative abundance of *Actinophytocola*, *Actinoplanes*, *Bacillus*, and *Russula* were significantly increased in healthy *C. coggygria* roots, while *Acidovorax*, *Microbacterium*, *Pseudomonas*, *Spingomonas*, *Diaporthe*, *Fusarium*, and *Morchella* were significantly enriched in diseased roots (Appendix A).

### 3.6. Co-Occurrence Networks of the Root Endophytic Microbiome

To investigate the variability in the key taxa and their interactions in response to Verticillium wilt occurrence, we analyzed the co-occurrence networks of bacterial and fungal communities in Verticillium wilt-affected and healthy *C. coggygria*. In the bacterial co-occurrence networks of healthy roots, *Bacillus*, *Solirubrobacter*, *Microbacterium*, *Mycobacterium*, and *Steroidobacter* were closely related to other bacterial genera, and their degrees of connection were the highest among all the nodes (Figure 6A). *Acidovorax*, *Pantoea*, *Sphingomonas*, *Microbacterium*, and *Bacillus* were the key taxa with high degrees of connection in the bacterial co-occurrence networks of diseased roots (Figure 6B).

For fungal communities, *Chrysosporium*, *Alternaria*, *Solicoccozyma*, and *Plectosphaerella* were the key taxa in the healthy co-occurrence networks (Figure 6C), and *Fusarium*, *Dictyocheirospora*, *Neocosmospora*, *Morchella*, and *Didymella* were the hub taxa in the diseased co-occurrence networks (Figure 6D). For bacterial network complexity, the degrees, degree centrality, closeness centrality, and network density of healthy plants were higher than those of diseased plants, although not significantly (Appendix A); however, those indices in the fungal networks of diseased plants were significantly increased compared with healthy plants. We also found that the fungal networks in the healthy plants had higher positive correlation ratios, while the fungal networks of diseased plants were more complex and denser with more edges, and the negative correlation ratios were higher compared with those of healthy plants (Appendix A).

## 4. Discussion

Verticillium wilt caused by *V. dahliae* is a typical soil-borne fungal disease, which is difficult to control with conventional methods [19]. Plant microbiomes play vital roles in plant health and disease resistance, and previous studies have demonstrated significant changes in microbial community diversity and structure in response to disease incidence [20]. Endophytic microbiome variability in response to Verticillium wilt occurrence has been reported on some plants such as cotton [11] and olive [21]. Zeng et al. [11] found that Verticillium wilt-affected samples had lower bacterial diversity when comparing the difference between infected and non-symptomatic cotton plants. In this study, the shifts in microbial diversity, community composition, dominant species, and co-occurrence networks of endophytic microbial communities were investigated between Verticillium wilt-affected and healthy *C. coggygria*. Our findings indicated that after the onset of Verticillium wilt, the bacterial diversity, as examined in Sobs, Chao, Ace, Shannon, and Invsimpson indices, significantly decreased, while the fungal diversity increased.

In this study, the endophytic bacteria were mainly composed of Proteobacteria, Actinobacteriota, Firmicutes, Myxococcota, and Chloroflexi, and their relative abundance differed significantly between Verticillium wilt-affected and healthy root samples. We also found that Actinobacteriota were distinctly less present in disease roots. Some species in Actinobacteriota have been reported to produce antibiotics and cause pathogen inhibition, so their enrichment in relative abundance may be related to the health maintenance or disease resistance in plant growth [22]. Tian et al. [23] found that nematode-infected tomato roots were associated with a decrease in the abundance of the major endophytes—Streptomycetaceae and *Pseudomonas*. Members of these groups were known to produce active compounds against plant pathogens. However, in this study, *Pseudomonas* was highly relatively abundant in diseased plants, and these might be non-antifungal strains or there might have been recruitment of other beneficial bacteria in response to *V. dahliae* infection [24]. In addition, Yang et al. [25] also found a higher abundance of *Pseudomonas* in rice leaves infected with bacterial blight. Genetic variations in plants may affect functional traits; therefore, the variability in microbial community and composition may have been influenced by both the health status and genotypes of plants [26]. Previous studies have found the significant enrichment of *Fusarium* in bacterial blight-infected rice leaves, and some species in this genus are known as major plant pathogens [25]. Tang et al. [27] also found *Fusarium* from the petals of rhododendron, which was associated with blossom rot disease. In this study, the relative abundance of *Fusarium* was increased in the roots of Verticillium wilt-affected plants, and this genus also displayed an important role in the complex co-occurrence networks.

The co-occurrence network of the bacterial community in healthy *C. coggygria* had more edges, a higher network density, and positive correlations, suggesting a more stable network with closer relationships formed within the community of healthy plants, which may have played an important role in resisting *V. dahliae* infection. Gao et al. [24] also found that the bacterial network in healthy pepper plants was more complex than in *Fusarium* wilt-affected plants. Mutually negative interactions in networks, often referred to as ecological competition, can improve microbiome stability by dampening the destabilizing effects of cooperation [28]. In this study, the fungal co-occurrence network of Verticillium wilt-affected plants had a higher network density and stronger species interactions, especially negative correlations, which may be related to the occurrence of and increase in the Verticillium wilt disease.

Some endophytic bacteria such as *Bacillus* and *Pseudomonas* in healthy plants were potentially beneficial to plants with antibacterial and plant growth-promoting activities [29]. Cui et al. [30] found that the *Bacillus* from healthy potato tubers have indole-3-acetic acid production and nitrogen fixation. Sallam et al. [31] found that the endophytic bacteria can secrete cell wall-degrading substances and prevent colonization by pathogenic bacteria, which may provide ways for biological disease control based on microbiome analysis and beneficial bacteria evaluation [32]. In this research, we analyzed the endophytic microbial diversity, community composition, dominant species occurrence, and the co-occurrence networks of *C. coggygria* under Verticillium wilt-affected and healthy conditions. Subsequent work will focus on the excavation and evaluation of beneficial bacterial strains to provide biocontrol resources and establish a theoretical basis for Verticillium wilt management in *C. coggygria*.

## 5. Conclusions

In summary, this study provided a comprehensive analysis of the root endophytic microbiome variability in Verticillium wilt-affected and healthy *C. coggygria*. The profile of the endophytic microbiome showed prominent differences in microbial taxa and abundance across different health conditions. These results contribute to our understanding of the relationship between endophytic microbiomes and Verticillium wilt occurrence. In the future, we can start by uncovering strains with growth promotion potential and disease control abilities to manage the occurrence of diseases.

## Figures and Tables

**Figure 1 jof-10-00792-f001:**
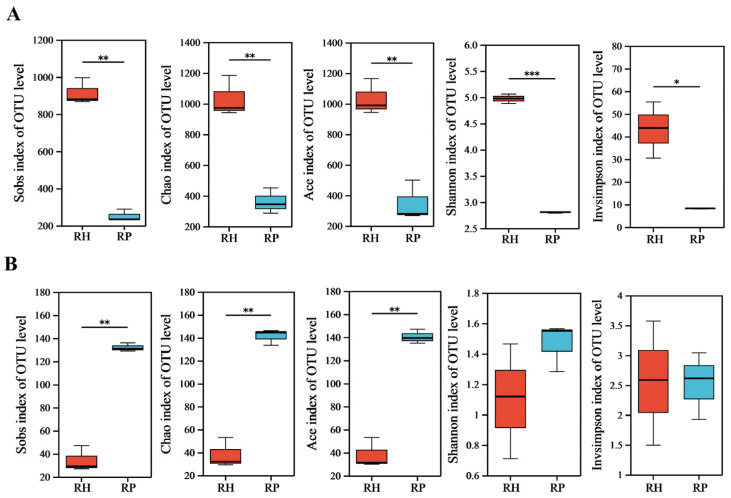
The alpha diversity indices of the root endophytic microbiome in Verticillium wilt-affected and healthy *C. coggygria* plants. (**A**) Bacterial community; (**B**) fungal community. RP represents the Verticillium wilt-affected roots; RH represents healthy roots. * indicates significant differences at a certain significance level (for example, *p* < 0.05); ** indicates significant differences at a higher significance level (for example, *p* < 0.01); *** indicates significant differences at a very high significance level (for example, *p* < 0.001).

**Figure 2 jof-10-00792-f002:**
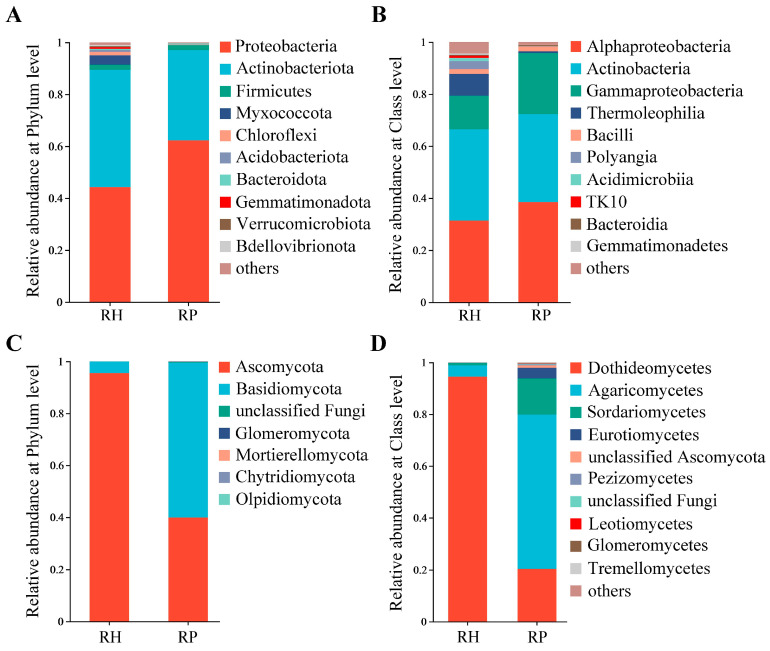
Community composition of the root endophytic microbiome in Verticillium wilt-affected and healthy *C. coggygria*. (**A**) Bacterial phylum level; (**B**) bacterial class level; (**C**) fungal phylum level; (**D**) fungal class level. RP represents the Verticillium wilt-affected roots; RH represents healthy roots.

**Figure 3 jof-10-00792-f003:**
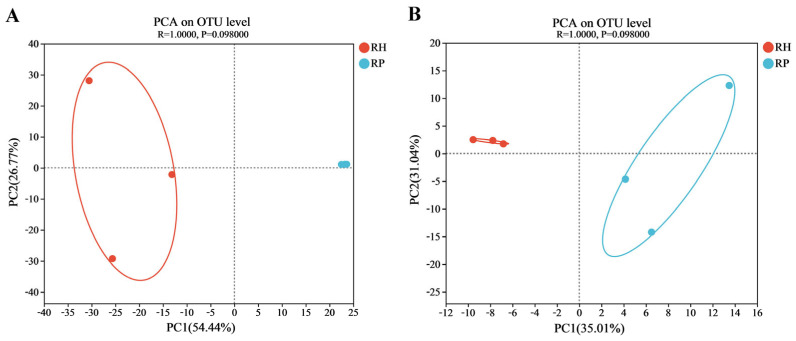
PCA of the root endophytic microbiome in Verticillium wilt-affected and healthy *C. coggygria* at the OTU level. The PC coordinate axis in the principal component analysis (PCA) graph denoted the difference in microbial composition. (**A**) PCA of the bacterial community; (**B**) PCA of the fungal community. RP represents the Verticillium wilt-affected roots; RH represents healthy roots.

**Figure 4 jof-10-00792-f004:**
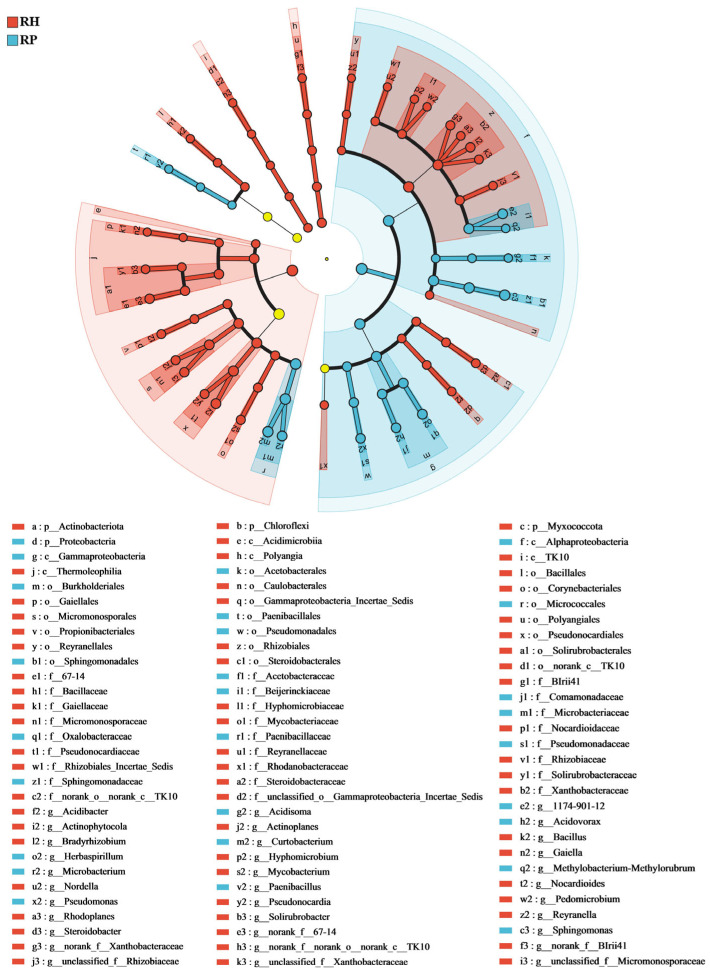
LEfSe analysis of the root endophytic bacterial of the Verticillium wilt-affected and healthy *C. coggygria*. The histogram of the LDA score calculates the differently abundant microbes among different health states, with a threshold of 3.5. Light yellow nodes represents microbial groups that show no significant differences among different groups or have no significant influence on the differences between groups. RP represents the Verticillium wilt-affected roots; RH represents healthy roots.

**Figure 5 jof-10-00792-f005:**
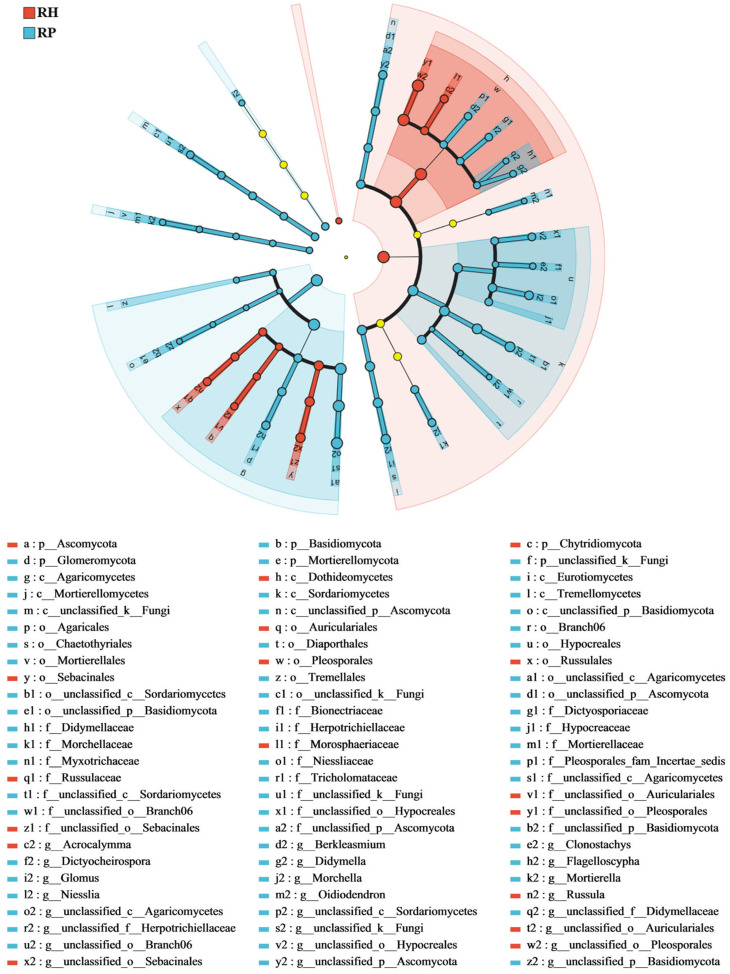
LEfSe analysis of the root endophytic fungal of the Verticillium wilt-affected and healthy *C. coggygria*. The histogram of the LDA score shows the differently abundant microbe among different health states, with a threshold of 3.0. Light yellow nodes represents microbial groups that show no significant differences among different groups or have no significant influence on the differences between groups. RP represents the Verticillium wilt-affected roots; RH represents healthy roots.

**Figure 6 jof-10-00792-f006:**
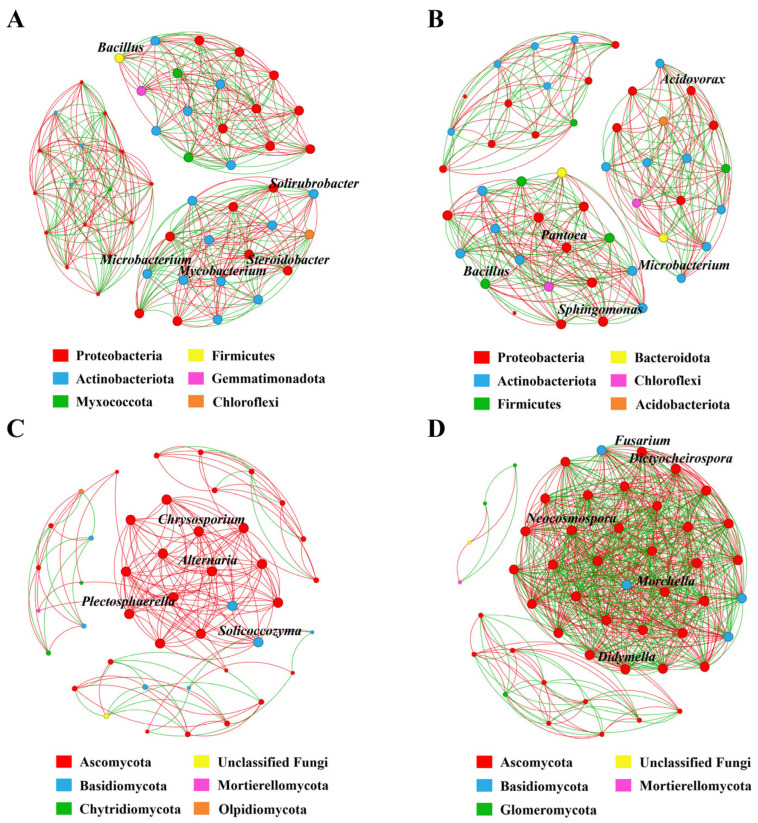
Co-occurrence networks of microbial communities of Verticillium wilt-affected and healthy *C. coggygria* at the genus level (Spearman > 0.8, *p* < 0.05). The size of the node shows the abundance of the genus, and the different colors indicate the corresponding taxonomic assignment at the phylum level. The edge color represents positive (red) and negative (green) correlations. (**A**) Bacterial networks from healthy plants, (**B**) bacterial networks from Verticillium wilt-affected plants, (**C**) fungal networks from healthy plants, and (**D**) fungal networks from Verticillium wilt-affected plants.

## Data Availability

The raw data were uploaded to the NCBI Sequence Read Archive with the submission accession numbers SAMN40539537-40539542 and SAMN40539821-40539826 under BioProject ID PRJNA1089384 and ID PRJNA1089391 for the bacterial and fungal sequences, respectively.

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
