# Peer review of "Response of the Endophytic Microbiome in Cotinus coggygria Roots to Verticillium Wilt Infection"

_jof, 2024, doi:10.3390/jof10110792_

Round 1
Reviewer 1 Report
The paper entitled “Response of the endophytic microbiome in Cotinus coggygria roots to Verticillium wilt infection” is an interesting work about a poorly investigated plant. The topic is clearly presented, even if introduction is quite brief and few details are provided. M&M are adequate, and the description is clear, even if some details about asymptomatic and symptomatic plants are mandatory. Results are significant and discussion is not speculative. Thus, I suggest publishing the article after correcting some minor parts.
L16-17: it seems that 2-infected conditions are evaluated, symptomatic and asymptomatic plants, and no proper control (healthy ones)
L25: is not really linked to V. dahlia infections, since both plant groups are infected, but due to the progress of the disease/showing of symptoms.
L41: besides Verticillium symptoms are generally similar, I’d like to know more details about the symptoms (o the whole plant) caused on Cotinus, because it is a plant poorly known outside China ed Asia.
L45-47: this citation is not really focused, because not linked to the topic neither of general interest about the role of beneficial microorganisms in plant protection.
L68: a key information is missing: what about plants details? Are the plants of the same age? Are the plants of the same genotype? How was it assessed? Without these details, differences found cannot be relevant.
L67-68 and 76-89: besides sequencing, a proper assessment of V. dahlia assessment with specific primers must be report. It is unusual carried out sequencing on groups of plant which should be infected by a pathogen but not testing it. Furthermore, we miss two fundamental data in order to differentiate the two groups: 1) how are symptoms evaluated? Are the symptomatic plants homogeneous in terms of symptoms? Are asymptomatic plants totally asymptomatic? 2) Is V. dahlie present in asymptomatic plants (as I think)? It is not really clear from the text. If so, a relative quantification of fungal presence is needed to compare the two plant groups. Moreover, a better clarification of objective should be provided (eg. the aim is to evaluate not the microbiota change cause by V. dahlia presence, but to the disease occurrence).
L132 and Fig. 2: it is not really clear why (and how) 3 different RH or RP are reported. In M&M, plants are 15 for each group, so I expected comparison among a mean-RH with a mean-RP (like Fig. 1), not just 3 plants (or 3 sub-group, I really miss this part in M&M).
Author Response
Responses to the Editor’s and Reviewers’ Comments
We thank the editor and reviewers for the constructive comments on our manuscript (jof-3259377) entitled “Response of the endophytic microbiome in Cotinus coggygria roots to Verticillium wilt infection” All the suggestions have been seriously considered and a revision has been carried out. The revised parts were highlighted in red text. The point-by-point responses to the comments are given as follows.
Reviewers 1’ Comments
Comments 1:[L16-17: it seems that 2-infected conditions are evaluated, symptomatic and asymptomatic plants, and no proper control (healthy ones)]
Response 1: Thank you for your suggestion. In this study, we have evaluated C. coggygria plants under two conditions: those affected by Verticillium wilt and those in a healthy state. C. coggygria affected by Verticillium wilt refers to the plant samples that exhibit distinct symptoms of Verticillium wilt, such as 2/3 of the leaves of the stems wilt or turn yellow. Moreover, these samples also show positive for pathogen detection using both universal primers and specific primers of V. dahliae. C. coggygria under healthy conditions means the plant samples showing no wilting symptoms and negative for pathogen detection.
We agree with that the word “Asymptomatic” in the original manuscript was not appropriate to describe C. coggygria plants that were not affected by Verticillium wilt. Therefore, we have used “healthy” to describe plants of no wilting symptoms and negative for pathogen detection instead of “asymptomatic” throughout the text and supplemented the description of the Verticillium wilt-affected and healthy plants in the revised manuscript (L80-85).
Comments 2:[ L25: is not really linked to V. dahlia infections, since both plant groups are infected, but due to the progress of the disease/showing of symptoms.]
Response 2: Thank you for your suggestion. In this study, we have evaluated C. coggygria plants under two conditions: those affected by Verticillium wilt and those in a healthy state. C. coggygria affected by Verticillium wilt refers to the plant samples that exhibit distinct symptoms of Verticillium wilt, such as 2/3 of the leaves of the stems wilt or turn yellow. Moreover, these samples also show positive for pathogen detection using both universal primers and V. dahliae specific primers. C. coggygria under healthy conditions means the plant samples showing no wilting symptoms and negative for pathogen detection. We have used “healthy” to describe plants of no wilting symptoms and negative for pathogen detection instead of “asymptomatic” throughout the text.
Co-occurrence networks revealed that the fungal network of Verticillium wilt-affected plants was denser with more negative interactions, compared with those of the healthy plants, so we deduced that this “may be relevant to functional changes from reciprocity to competition in the microbial community, in response to V. dahliae infection.”
Comments 3:[ L41: besides Verticillium symptoms are generally similar, I’d like to know more details about the symptoms (o the whole plant) caused on Cotinus, because it is a plant poorly known outside China ed Asia.]
Response 3: Thank you for your suggestion. The typical symptoms of Verticillium wilt on C. coggygria comprise yellowing, wilting and curling of leaf surfaces, withering and drying up of inflorescences, discoloration of vascular tissues, retarded growth, withering of individual branches, and ultimately the demise of the entire tree. We have supplemented this description in the revised manuscript. (L41-44)
Comments 4:[ L45-47: this citation is not really focused, because not linked to the topic neither of general interest about the role of beneficial microorganisms in plant protection.]
Response 4: Thank you for your suggestion. we have deleted this citation in the revised manuscript according to suggestion. (L49-52)
Comments 5:[ L68: a key information is missing: what about plants details? Are the plants of the same age? Are the plants of the same genotype? How was it assessed? Without these details, differences found cannot be relevant.]
Response 5: Thank you for your suggestion. We have supplemented “All the C. coggygria var. pubescens were about 2.5 to 3 meters in height, with crown diameter of 2.5 to 3 meters and trunk diameter of 10 centimeters.” in the revised manuscript (L72-73)
Comments 6:[ L67-68 and 76-89: besides sequencing, a proper assessment of V. dahlia assessment with specific primers must be report. It is unusual carried out sequencing on groups of plant which should be infected by a pathogen but not testing it. Furthermore, we miss two fundamental data in order to differentiate the two groups: 1) how are symptoms evaluated? Are the symptomatic plants homogeneous in terms of symptoms? Are asymptomatic plants totally asymptomatic? 2) Is V. dahliae present in asymptomatic plants (as I think)? It is not really clear from the text. If so, a relative quantification of fungal presence is needed to compare the two plant groups. Moreover, a better clarification of objective should be provided (eg. the aim is to evaluate not the microbiota change cause by V. dahlia presence, but to the disease occurrence).]
Response 6: Thank you for your suggestion.
We graded the disease severity of Verticillium wilt from 0 to 4 based on the reference of “Wang et al., 2013”. Level 0: no wilting, Level 1: <5 leaves turn yellow or wilt, Level 2: 5-10 leaves turn yellow or wilt, Level 3: leaves on 2/3 stems turn yellow or wilt, Level 4: >85% of the leaves wilted, fell off, or died entirely. We used two plant groups, the Verticillium wilt-affected and healthy plants.
Firstly, we investigated the selected C. coggygria plants at the sampling site based on the levels of disease severity. C. coggygria plants with disease severity of Verticillium wilt rated as grade 3-4 level and positive for pathogen detection were classified as Verticillium wilt-affected plants. Secondly, we also detected the pathogen using universal primers and specific primers for V. dahliae. Those without wilting symptoms and negative for pathogen detection were considered healthy plants. We have supplemented these descriptions in the revised manuscript according to the reviewer’s suggestions. (L73-85)
Comments 7:[ L132 and Fig. 2: it is not really clear why (and how) 3 different RH or RP are reported. In M&M, plants are 15 for each group, so I expected comparison among a mean-RH with a mean-RP (like Fig. 1), not just 3 plants (or 3 sub-group, I really miss this part in M&M).]
Response 7: Thank you for your suggestion. A total of 15 Verticillium wilt-affected and 15 healthy plants of C. coggygria var. pubescens was selected to obtain plant samples, with three replications and each replication five plants. We have supplemented the description of sample grouping in the revised manuscript.(L83-85)
At the same time, we have revised Fig. 2 to show the average RH and the average RP according to the reviewer’s suggestion.
Reviewer 2 Report
As part of plant microbiome endophytes have the potential to help enhance the plant's defense against plant pathogens. Major advances in genome sequencing have led to a better understanding of the microbiome structure and function. A shift in the microbiome balance is linked to a broad range of plant diseases. This understanding may lead to potential opportunities to develop next-generation microbiome-based disease management options and diagnostic biomarkers. However, our understanding is limited given the nature of the plant microbiome and its complex and multidirectional interactions.The manuscript titled “Response of the endophytic microbiome in Cotinus coggygria roots to Verticillium wilt infection” by Yanli Cheng et al. describes analysis of diversity of endophytic microbial communities of Verticillium wilt-affected and asymptomatic Cotinus coggygria roots using high-throughput sequencing technology.
Major concern
Page 1, Lines 67 - 69
Typical Verticillium wilt-affected and asymptomatic plants of C. coggygria were selected to obtain 15 samples of each in a 400 m2 area.
It is not clear whether the symptomatic plant samples were processed for laboratory isolation and positive confirmation of verticillium fungus. This is a very fundamental requirement.
The manuscript is well-written and presents an interesting and careful bioinformatic analysis and visualization of endophytic microbiome of Verticillium wilt-affected and asymptomatic Cotinus coggygria roots.
Please refer to major comments
Author Response
Responses to the Editor’s and Reviewers’ Comments
We thank the editor and reviewers for the constructive comments on our manuscript (jof-3259377) entitled “Response of the endophytic microbiome in Cotinus coggygria roots to Verticillium wilt infection” All the suggestions have been seriously considered and a revision has been carried out. The revised parts were highlighted in red text. The point-by-point responses to the comments are given as follows.
Reviewers 2’ Comments
Comments 1:[Lines67–69: Typical Verticillium wilt-affected and asymptomatic plants of C. coggygria were selected to obtain 15 samples of each in a 400 m2 area. It is not clear whether the symptomatic plant samples were processed for laboratory isolation and positive confirmation of verticillium fungus. This is a very fundamental requirement.]
Response 1:Thank you for your suggestion. All the plant samples were processed for laboratory isolation and confirmation of V. dahliae. In this study, C. coggygria plants with disease severity of Verticillium wilt rated as grade 3-4 level and positive for pathogen detection were classified as Verticillium wilt-affected plants, and those without wilting symptoms and negative for pathogen detection were considered healthy plants. We have supplemented these descriptions in the revised manuscript. (L73-83)

Round 2
Reviewer 2 Report
No comments at this stage
No comments at this stage